# Evidence-R1: Fine-Grained Evidence-Driven Explicit Reasoning and Implicit Reflection for Enhancing RAG Explainability via Reinforcement Learning

## Abstract

Although Retrieval-Augmented Generation (RAG) has effectively mitigated the factual hallucination inherent in large language models (LLMs) by integrating external retrieved knowledge, LLMs still struggle with explainability and traceability. Existing research mainly focuses on generating responses with in-line citations, which can serve as evidence with factualness and verifiability. However, conducting fine-grained verification of such citations and mitigating citation errors remain significant challenges. To address this issue, we propose **Evidence-R1**, a novel RAG generator framework which drives explicit reasoning and implicit reflection based on sentence-level evidence. Specifically, explicit reasoning is defined as a reasoning process that strictly requires explicitly inferring answers from cited sentence-level evidence, while implicit reflection serves as an internal self-checking process that evaluates whether such answers are supported by the evidence through a special token, called $Sup$. Nevertheless, this approach occasionally introduces asymmetry in the sentence-level evidence relied upon by the two processes. To tackle this, we introduce Multi-reward Dependence-aware Alignment (**MRDAA**), a multi-rule tree reward mechanism that enhances the consistency between the two processes through reinforcement learning. Experimental results on the ALCE benchmark dataset demonstrate that Evidence-R1 outperforms existing state-of-the-art models in citation precision, even surpassing ChatGPT. Furthermore, by implementing fine-grained verification, Evidence-R1 has achieved significant improvements in interpretability and traceability.(https://anonymous.4open.science/r/Evidence-R1-1993699F/)

## 1 Introduction

Large language models (LLMs) Brown et al. (2020); Achiam et al. (2023); Zhao et al. (2023) have demonstrated state-of-the-art performance across a wide range of tasks, including question answering Zhao et al. (2023), text generation Li et al. (2024), and code synthesis Piterbarg et al. (2024). Despite these advanced capabilities, LLMs still struggle with factual hallucinations Rawte et al. (2023); Menick et al. (2022), which involve generating fabricated facts or unfaithful content. This issue significantly undermines the reliability and credibility of LLMs. By retrieving and incorporating relevant external knowledge, Retrieval-Augmented Generation (RAG) methods effectively enhance generated outputs of LLMs and mitigate the hallucination, especially in knowledge-intensive tasks Ram et al. (2023); Asai et al. (2023). However, as illustrated in the far left section of Figure 1, LLMs occasionally produce unsupported or contradictory statements (highlighted in red) to the retrieved passages, and such issues are often difficult to verify whether generated answers originate from external retrieval knowledge or the model itself. Therefore, it is especially crucial to provide source-tracing evidence for answers in terms of how they can be explained and tracked. some studies obtain source-tracing evidence via post-hoc retrieval Gao et al. (2022), where LLMs first generate an initial response and then retrieve the most relevant evidence to support it. Unfortunately, these methods suffer from a critical limitation: the generation and retrieval processes operate in isolation, with little to no synergy between them. This disconnection ultimately leads to suboptimal citation quality in the final results. In contrast, another line of research Gao et al. (2023); Asai et al. (2024);

Huang et al. (2024) mainly focuses on generating responses with in-line citations via in-context learning that act as reliable, verifiable evidence to support the facts in the answers. As illustrated in the middle section of Figure 1, although this paradigm effectively mitigates the answer hallucination, it is difficult for users to accurately assess whether the model has incurred misunderstandings or engaged in over-inference during the information integration process. Therefore, conducting fine-grained verification of these citations and addressing citation errors (highlighted in red) remain significant emerging challenges.

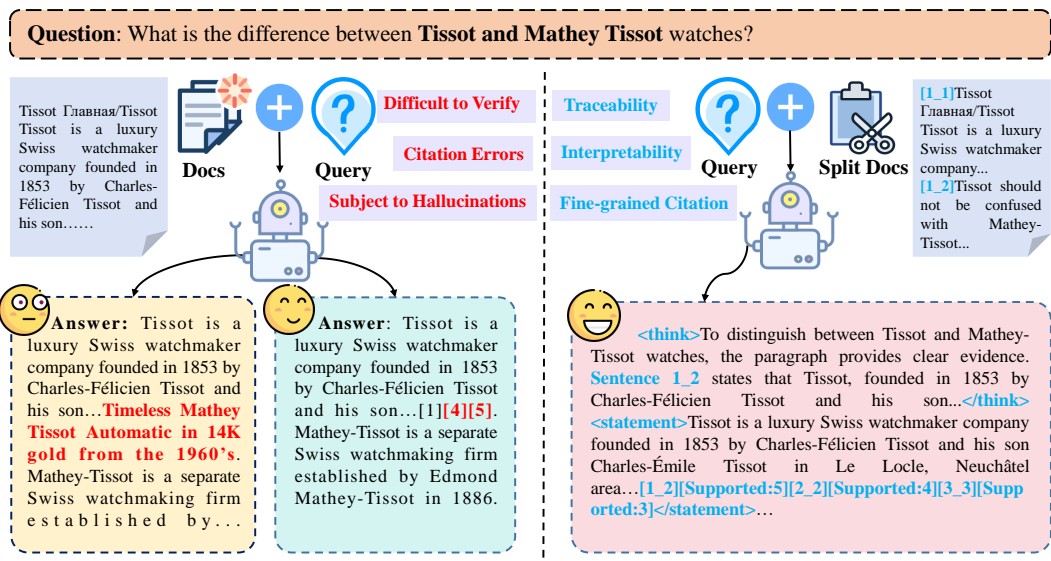

Figure 1: Compared with current RAG, Evidence-R1 first provides a reasoning process based on cited sentence-level evidence, and then evaluates how well cited evidence supports the answer. This ensures interpretability, traceability and more accurate citations.

To address these issues, this paper proposes **Evidence-R1**, a novel RAG generator framework designed to implement fine-grained verification through a dual-process mechanism: explicit reasoning and implicit reflection on sentence-level evidence. As shown in the right section of Figure 1, LLMs firstly generate answers through explicitly reasoning based on cited sentence-level evidence, as opposed to relying on coarse-grained document-level information. And then, the implicit reflection serves as an internal self-checking mechanism to evaluate whether each statement of the answers is supported by the cited sentence-level evidence. This evaluation is operationalized via a special token, denoted as $Sup$, which employs a five-point scale (ranging from 1, indicating the lowest level of support, to 5, representing the highest level of support). However, obtaining such data of fine-grained verification for supervised fine-tuning is both difficult and costly. Thus, we propose an automatic data generation pipeline that leverages ChatGPT Welsby & Cheung (2023) to synthesize high-quality thinking processes enclosed within $< think >$ and $< /think >$ tokens, along with cited answers and reflection tokens wrapped in $< statement >$ and $< /statement >$ tokens, as illustrated in Figure 1. In addition, Evidence-R1 occasionally introduces asymmetry in the sentence-level evidence relied upon by above two processes, explicit reasoning and implicit reflection. To tackle this, we introduce Multi-reward Dependence-aware Alignment (**MRDAA**), a multi-rule tree reward mechanism designed to handle the inherent interdependence among multiple rewards. In this mechanism, the total reward is a weighted sum of all nodes' rewards in the tree while each node's reward is the cumulative product of the probabilities of all its ancestor nodes and its own reward. This design effectively enhances the consistency between the two processes via reinforcement learning. Finally, to ensure high-quality sentence-level citation evidence, we apply a citation filtering process guided by the reflective token to enhance the accuracy and reliability of the evidence. Experiments on the ALCE benchmark dataset demonstrate the effectiveness of our method, which not only outperforms state-of-the-art models (including ChatGPT) in citation precision, but also achieves significant improvements in interpretability and traceability, thereby enhancing RAG explainability.

## 2 RELATED WORK

**Retrieval-Augmented Generation.** Retrieval-Augmented Generation (RAG) has achieved remarkable progress in bridging the knowledge gap between models and the real world, enhancing the performance of LLM-generated content through the integration of retrieved relevant informationFan et al. (2024); Sawarkar et al. (2024); Tan et al. (2024). However, prior methods suffer from inconsistencies between generated answers and retrieved documents, while also lacking in attribution and verifiability of the generated content. Recently, extensive research has focused on enhancing the citation capability of answers. Ji et al. (2024) utilized Chain-of-Thought (CoT) to guide LLMs in their ability to synthesize correct answers from multiple documents and to correctly cite these documents. Huang et al. (2024) designed the FRONT method, which improves citation quality and fine-grained verification by grounding the model output in fine-grained supporting citations to guide generation. Xia et al. (2025) proposed a novel self-inference framework that leverages LLMs' own generated inference trajectories to enhance their reliability and traceability. Nevertheless, the citations in the generated answers of these methods remain coarse-grained, precluding rapid and accurate sentence-level attribution. In contrast, our proposed method enables sentence-level citations for verification through explicit reasoning and implicit reflection tokens, further enhancing the quality and reliability of answer generation.

## 3 EVIDENCE-R1

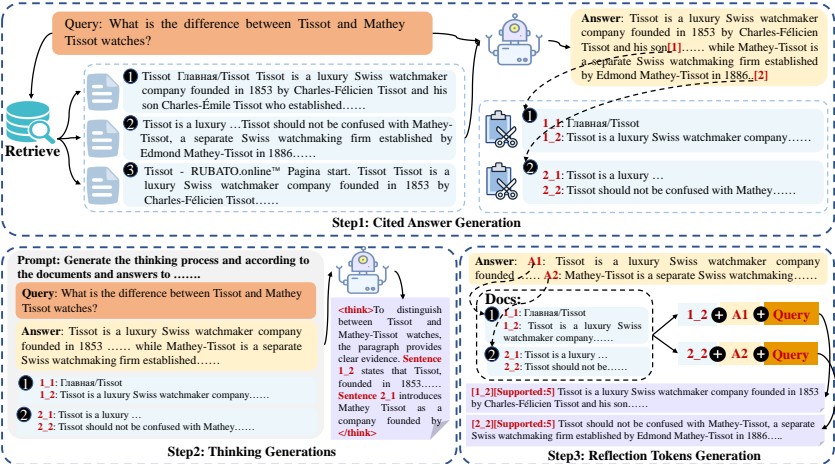

Figure 2: Overview of the data generation pipeline which comprising three components—Cited Answer Generation, Thinking Generation, and Reflection Tokens Generation. Cited Answer Generation is to generate answers with citations from retrieved documents which can be split into sentences. Thinking Generation produces a reasoning process, strictly using sentence-level evidence from cited passages. And Reflection Tokens Generation creates the special token $Sup$ to evaluate how well the cited evidence supports the answer.

### 3.1 PROBLEM FORMALIZATION

Given a natural language question $q$ and a corpus of $n$ retrieved reference passages $D = \{d_1, d_2, ..., d_n\}$, our Evidence-R1 generates a structured response that includes a reasoning process $Eot$ derived from fine-grained sentence-level evidence and an answer $a = \{s_1, s_2, ..., s_m\}$ accompanied by reflective evaluation based on cited sentence-level evidence as follows:

$$Eot, \sum_{i=1}^{m} s_i (\sum_{j=1}^{k_i} c_{ji} Sup_{ji}) = LLM(q, D) \tag{1}$$

where $LLM$ is the large language model used, $s_i$ refers to the $i$-th sentence of $a$ and $k_i$ is the number of the sentence-level evidence list $C_i \subset D$ related to $s_i$. In addition, For each $c_{ji} \in C_i$, We conduct a reflective evaluation based on $q$ and $s_i$, with a reflection token: $Sup_{ji}$, referring to a five-scale evaluation (1 is the lowest and 5 is the highest) whether $s_i$ is fully supported by $c_{ji}$.

### 3.2 DATA GENERATION

Manual annotation of high-quality training data is typically both labor-intensive and costly, while state-of-the-art large language models (LLMs) like ChatGPT can be effectively leveraged to synthesize such annotations. Therefore, we propose a pipeline for automatically synthesizing training data, which comprises three core components: cited answer generation, thinking generation, and reflection tokens generation, as outlined in Figure 2. See Appendix B.2 for more details.

#### 3.2.1 CITED ANSWER GENERATION.

Similar to FRONT Huang et al. (2024), to simulate real-world information retrieval scenarios, we select questions from the AQuAMuSe Kulkarni et al. (2020) dataset derived from the Natural Question (NQ) Kwiatkowski et al. (2019) dataset, which contains real Google search queries covering diverse question types and answer length requirements. Using the pre-processed Sphere Piktus et al. (2021) corpus as a proxy for web search indexes, for each sampled question, we first retrieve the top 100 relevant documents from the Sphere corpus via sparse retrieval. And these documents are then re-ranked by RankVicuna Pradeep et al. (2023), resulting in the top 5 most relevant documents per query. Finally, we prompt ChatGPT to generate answers where each sentence cites relevant passages in the format of [1][2], based on the given question and the top 5 retrieved documents.

#### 3.2.2 THINKING AND REFLECTION TOKENS GENERATION.

Based on the answers generated above, we first extract each passage cited in the answers, split it into sentences using NLTK Bird (2006), and form a sentence-level evidence passage $E$. Next, we employ ChatGPT to generate a sentence-level thinking process based on the given question, answers and $E$. Specifically, the reasoning process is strictly required to reason the answers based on fine-grained sentence evidence, with relevant sentences referenced only via sentence labels in the format of "sentence 1_1, 1_2". Meanwhile, we split the answers into sentences $A$ using NLTK. For each sentence $a_i$ in $A$, we prompt ChatGPT to extract sentence-level evidence $S$ from $E$ based on the given question and $a_i$. Subsequently, for each sentence $e_j$ in $S$, we generate the reflection token ($Sup_{ji}$) based on the given question and current answer $a_i$. In more detail, we evaluate whether $a_i$ is fully supported by the information provided in $e_j$ with a five-scale evaluation $Sup_{ji}$ (1 is the lowest and 5 is the highest).

### 3.3 TWO-STAGE TRAINING AND CITATION FILTERING

In this section, as illustrated in Figure 3, we describe the implementation details of Evidence-R1, which consists of two modules: Two-Stage Training and Citation Filtering.

In the Two-Stage Training module, we first introduce **EoT (Evidence-of-Thought) and Reflection Token Generation**, enabling the model to initially acquire explicit reasoning capabilities and implicit reflection abilities. Concurrently, to address interdependencies in rule-based reward mechanisms, we propose **Multi-reward Dependence-aware Alignment** to effectively enhance the model's explicit reasoning and implicit reflection. The **Citation Filtering** module filters the already cited sentence-level evidence based on reflection tokens.

#### 3.3.1 EOT AND REFLECTION TOKEN GENERATION.

To enable LLM to possess explicit reasoning and implicit reflection capabilities based on fine-grained sentence-level evidence such as Equation 1, we propose EoT and Reflection Token Generation (**ERTG**) through supervised fine-tuning using the aforementioned synthetic data. Firstly, LLM is allowed to conduct explicit reasoning based on sentence-level evidence as follows:

$$Eot = <think>eot_1, ..., eot_i, eot_t</think> \tag{2}$$

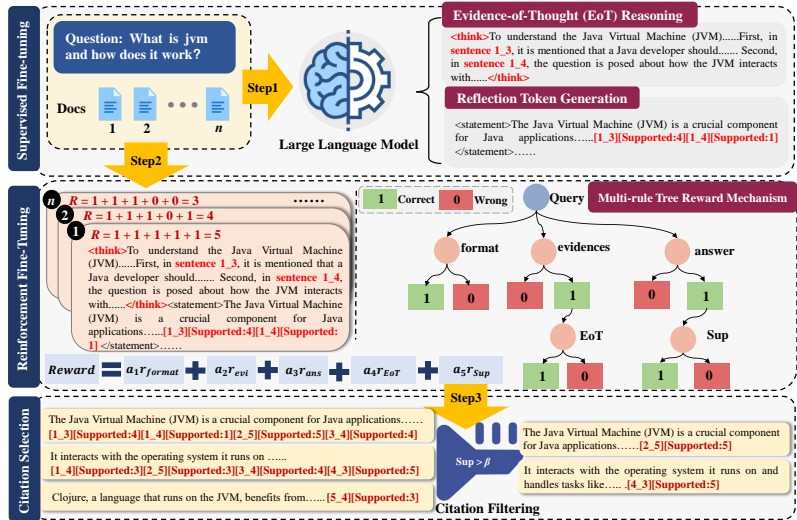

Figure 3: Overview of Evidence-R1. There are two Stage Training and Citation Filtering, which enable fine-grained verification through both explicit reasoning and implicit reflection based on sentence-level evidence. In the Two-Stage Training module, we enable the model to acquire explicit reasoning capabilities and implicit reflection abilities by supervised fine-tuning and reinforcement learning. Citation Filtering is used to select citations.

where the special tokens, $<think>$ and $</think>$, indicate the start and end of the explicit reasoning process while $eot_i$ is the reasoning process based on the sentence-level evidence $c_i$.

Secondly, LLM generates answers $a = \{s_1, s_2, ..., s_m\}$ and reflection tokens presented as:

$$A = \sum_{i=1}^{m} s_i (\sum_{j=1}^{k_i} c_{ji} Sup_{ji}) \tag{3}$$

where for each sentence $s_i$ in $a$, the corresponding sentence-level evidence $c_{ji}$ is cited, and an implicit reflective token ($Sup_{ji}$) is conducted to determine whether $s_i$ is fully supported by the information provided in $c_{ji}$.

Thus, the training loss is formulated as:

$$L = -\sum_{i=1}^{N} logP(y_i|q_i, D_i; \theta) \tag{4}$$

where $y_i$ is the combined output of $Eot$ and $A$ for each given question $q_i$ and retrieved reference passages $D_i$.

### 3.3.2 MULTI-REWARD DEPENDENCE-AWARE ALIGNMENT.

While ERTG equips LLM with foundational explicit reasoning and implicit reflection capacities, it occasionally induces asymmetry in their relied-upon sentence-level evidence. Such inconsistency detrimentally impacts overall model performance. To achieve this, a simple approach is to leverage reinforcement learning to enhance the consistency between explicit reasoning and implicit reflection. However, Unlike previous scenarios where multiple rewards operate independently, the multiple components of the results generated by this method are inherently interdependent. For instance, $Eot$ is an explicit reasoning process that includes cited sentence-level evidence, while the generation of reflection tokens is constrained by both the accuracy of the answer and the validity of the cited sentence-level evidence. So, we propose Multi-reward Dependence-aware Alignment (**MRDAA**), a multi-rule tree reward mechanism, to effectively enhance the consistency of sentence-level evidence and the model's performance in explicit reasoning and implicit reflection evaluation.

Currently, the GRPO Guo et al. (2025) method of DeepSeek-R1 has been employed to enhance model capabilities. This approach eliminates the need for an additional critic model by directly

comparing groups of diverse candidate outputs. Specifically, we leverage Group Relative Policy Optimization (**GRPO**) for MRDAA, to optimize the generation of explicit reasoning and implicit reflection. For instance, given a question $q$ and retrieved reference passages $D$, the old policy $\pi_{\theta_{old}}$ firstly generates distinct outputs $\{a_1, a_2, ..., a_G\}$ and then GRPO calculates the relative advantage $A_j$ for each output $a_j$ as follows:

$$A_j = \frac{r_j - \text{mean}\{r_1, r_2, \cdots, r_G\}}{\text{std}(\{r_1, r_2, \cdots, r_G\})} \tag{5}$$

where $r$ is the reward given by a multi-rule tree reward function, which we describe in detail later. Finally, GRPO optimizes the current policy $\pi_\theta$ by maximizing the following objective function:

$$
\begin{aligned}
\mathcal{J}_{\text{GRPO}}(\theta) = & E\left[q, \{a_i\}_{i=1}^G \sim \pi_{\theta_{\text{old}}}\right] \\
& \frac{1}{G}\sum_{i=1}^{G}\bigg(\min\big(\frac{\pi_\theta(a_i|q,D)}{\pi_{\theta_{\text{old}}}(a_i|q,D)}A_i, \\
& \text{clip}(\frac{\pi_\theta(a_i|q,D)}{\pi_{\theta_{\text{old}}}(a_i|q,D)}, 1-\varepsilon, 1+\varepsilon)A_i\big) \\
& - \beta\text{KL}\left(\pi_\theta\|\pi_{\text{ref}}\right)\bigg)
\end{aligned}
\tag{6}
$$

where $\varepsilon$ and $\beta$ are hyperparameters. $\text{clip}(\cdot)$ is employed to cap the ratio between the new and old policies, while $\text{KL}(\cdot)$ denotes the KL loss that penalizes how far the new policy $\pi_\theta$ differs from the reference policy $\pi_{\text{ref}}$ and effectively constrains the magnitude of policy updates to mitigate training instability.

Our method generates structured results composed of multiple components, specifically including explicit reasoning, answers, sentence citation, and implicit reflection tokens. Unlike previous scenarios where multiple reward mechanisms operate independently, these components are interdependently related. To address this, we propose a multi-rule tree reward mechanism (**MRTR**) to enable mutual learning and promotion among multiple rewards. The reward of this mechanism is equal to the weighted sum of all nodes' rewards in the tree, while the reward of a single node is the cumulative product of the probabilities of all its ancestor nodes and its own reward. As is shown in Figure 2, our MRTR is defined by the following equation:

$$
\begin{aligned}
r =& \alpha_1 r_{format} + \alpha_2 r_{evi} + \alpha_3 r_{ans} + \alpha_4 r_{Eot} + \alpha_5 r_{Sup} \\
=& \alpha_1 r_{format} + \alpha_2 r_{evi} + \alpha_3 r_{ans} + \alpha_4 \frac{\sum_{i=1}^{t} r_{eot_i|evi_i}}{t} + \alpha_5 \frac{\sum_{j=1}^{k} r_{Sup_j|ans_j}}{k} \\
=& \alpha_1 r_{format} + \alpha_2 r_{evi} + \alpha_3 r_{ans} + \alpha_4 \frac{\sum_{i=1}^{t} P(evi_i) r_{eot_i}}{t} + \alpha_5 \frac{\sum_{j=1}^{k} P(ans_j) r_{Sup_j}}{k}
\end{aligned}
\tag{7}
$$

where $\alpha_1, ..., \alpha_6$ are the weights of different rewards and $t, k$ are the numbers of sentence-level evidence in Evidence-of-Thought (EoT) Reasoning and Reflection Token Generation, respectively. $r_{format}$ is used to match the expected structured data. A reward of 1 is given if the output meets the requirements; otherwise, 0 is given. $r_{evi}$ is aimed to evaluate the correctness of the citation while $r_{ans}$ represents the accuracy of final answers. Since our main focus is on the citation quality, we simply set $r_{ans}$ to 1 here. In particular, $r_{Eot}$ and $r_{Sup}$ represent constrained rewards. $r_{eot_i|evi_i}$ constrains the explicit reasoning to include at least one sentence-level evidential source from the answer section. $r_{Sup_j|ans_j}$ is constrained by the accuracy of the current answer: the model's reflective evaluation of sentence evidence is only valid when the answer is accurate. $P(evi_i)$ and $P(ans_j)$ represent probabilities: $P(evi_i)$ refers to the probability of $evi_i$ cited, while $P(ans_j)$ is aimed to evaluate whether $ans_j$ is the answer for the current sentence-level evidence, where we simply use BLEU scores.

### 3.3.3 Citation Filtering.

To ensure high-quality sentence-level citation evidence, we apply a reflective token-guided citation filtering process, selecting citations with scores above the threshold to enhance their accuracy and reliability.

## 4 Experiments

### 4.1 Datasets and Settings

We evaluate our method (Evidence-R1) on the ALCE benchmark Gao et al. (2023), designed for automatic LLM's citation evaluation. The benchmark includes three QA datasets: ASQA Stelmakh et al. (2022), QAMPARI Amouyal et al. (2022), and ELI5 Fan et al. (2019), which are used to automatically evaluate the fluency, correctness, and citation quality of text generated by LLMs. ASQA is a long-form factoid dataset containing ambiguous questions from AmbigQA. These questions need multiple short answers to cover all aspects, while ASQA offers a long-form answer that incorporates all such short answers. ELI5 is a long-form QA dataset from the Reddit forum, which is suitable for open-domain long-form abstract QA tasks and requires long answers supported by multiple passages as evidence. QAMPARI is a factoid QA dataset built from Wikipedia, with answers taking the form of entity lists sourced from various passages. See Appendix B.1 for more details.

Following the ALCE benchmark, our evaluation focuses on two core dimensions: Citation Quality and Correctness. And given our primary focus on Citation Quality, we set $\alpha_1, ..., \alpha_6 = 1$ and $r_{ans} = 1$.

**Citation Quality.** Citation quality is measured by two core metrics: citation precision and citation recall. Moreover, to capture a comprehensive measure of citation quality, we also report the Citation F1 score, which is the harmonic mean of citation precision and recall.

**Correctness.** For ASQA, we calculate exact match recall (EM Rec.) by checking if the short answers are exact substrings in what the model generates. For ELI5, we use claim recall (Claim) to see whether the model's response entails the ground truth sub-claims. And For QAMPARI, we measure correctness with exact match precision (Prec.) and top-5 exact match recall (Rec.-5).

### 4.2 Baseline Models

We conduct a comparison of Evidence-R1 against three types of baselines:

**Prompting-based Methods:** In this setting, we feed the query and relevant retrieved documents into LLMs to generate an answer with in-line citations by in-context learning method. We evaluate the powerful closed-source model GPT-3.5-Turbo and the strong open-source pre-trained LLMs on LLaMA-2 series (LLaMA2-7B, LLaMA2-13B, LLaMA2-70B) and Mistral series spans from Mistral-7B Jiang et al. (2023) to Mistral-Mistral-8x7BMoE Jiang et al. (2024) as baseline models.

**Post-hoc Retrieval Methods:** We benchmark this baseline by utilizing the same models mentioned in prompting-based settings. First, we prompt LLMs to generate query-based answers, after which the model retrieves relevant documents by leveraging both the query and its generated response. Ultimately, GTR Ni et al. (2021) selects the top-scoring documents that contains the LLM-generated answer as attribution.

**Training-based Methods: Self-RAG** Asai et al. (2024) trains LLMs to adaptively retrieve information on demand and engage in self-reflection, thereby enhancing output quality and factuality. **FRONT** Huang et al. (2024) designs a LLM that first generates precise grounding and subsequently guides the generation of attributed answers, thereby mitigating hallucinations and boosting citation verifiability in outputs.

### 4.3 Main Results

Table 1 shows the performance comparisons in citation quality with different methods on the ALCE benchmark, only part of the answer is lost. Evidence-R1 outperforms all baseline methods in ci-

Table 1: Results on the ALCE benchmark. **Bold** numbers indicate the best performance, while _ indicates the second-best performance.

| Models | Model Size | ASQA Correctness EM Rec. | ASQA Citation Rec. | ASQA Citation Prec. | ASQA F1 | ELI5 Correctness Claim | ELI5 Citation Rec. | ELI5 Citation Prec. | ELI5 F1 | QAMPARI Correctness Rec.-5 | QAMPARI Correctness Prec. | QAMPARI Citation Rec. | QAMPARI Citation Prec. | QAMPARI F1 |
|---|---|---|---|---|---|---|---|---|---|---|---|---|---|---|
| *Prompting-based* | | | | | | | | | | | | | | |
| ChatGPT | - | 40.37 | 72.81 | 69.89 | 71.22 | 12.47 | 49.44 | 47.05 | 48.22 | 20.28 | 19.84 | 19.06 | 22.03 | 20.44 |
| LLaMA-2 | 7B | 24.32 | 17.24 | 17.67 | 17.55 | 4.53 | 3.92 | 5.38 | 4.54 | 12.56 | 11.32 | 6.03 | 6.35 | 6.19 |
| | 13B | 27.99 | 16.45 | 19.04 | 17.65 | 7.77 | 8.49 | 8.43 | 8.46 | 18.00 | 12.39 | 5.45 | 5.74 | 5.59 |
| | 70B | 31.53 | 44.18 | 44.79 | 44.48 | 10.43 | 23.75 | 22.43 | 23.07 | 18.50 | 14.79 | 10.10 | 10.50 | 10.30 |
| LLaMA-2-Chat | 7B | 29.93 | 55.99 | 51.66 | 53.74 | 12.47 | 19.90 | 15.48 | 17.41 | 17.96 | 19.74 | 9.58 | 9.68 | 9.63 |
| | 13B | 34.39 | 37.15 | 38.17 | 37.65 | 13.83 | 16.50 | 16.09 | 16.29 | 21.34 | 18.86 | 8.94 | 9.06 | 9.00 |
| | 70B | 41.24 | 60.19 | 61.16 | 60.67 | 13.30 | 36.63 | 36.63 | 36.63 | 22.62 | 18.04 | 13.49 | 13.98 | 13.73 |
| Mistral | 7B | 29.46 | 23.12 | 25.45 | 24.23 | 8.47 | 16.96 | 15.98 | 16.18 | 16.96 | 15.98 | 7.50 | 7.70 | 7.63 |
| | 8 × 7B | 36.30 | 32.72 | 34.49 | 33.58 | 10.43 | 26.11 | 25.09 | 25.59 | 18.18 | 15.63 | 9.72 | 10.26 | 9.95 |
| Mistral-Instruct | 7B | 38.57 | 64.90 | 59.67 | 62.18 | 11.07 | 49.25 | 42.69 | 45.74 | 17.52 | 21.29 | 17.56 | 18.53 | 18.03 |
| | 8 × 7B | **44.11** | 61.80 | 63.27 | 62.53 | 13.93 | 49.28 | 48.34 | 48.81 | 20.12 | 19.64 | 19.27 | 20.38 | 19.81 |
| *Post-hoc Retrieval* | | | | | | | | | | | | | | |
| ChatGPT | - | 37.68 | 27.11 | 27.05 | 27.08 | **18.77** | 14.55 | 14.55 | 14.55 | **25.14** | **22.85** | 12.29 | 12.29 | 12.29 |
| LLaMA-2-Chat | 70B | 29.68 | 24.51 | 24.51 | 24.51 | 16.03 | 12.93 | 12.93 | 12.93 | 17.90 | 14.45 | 9.05 | 9.05 | 9.05 |
| Mistral-Instruct | 8 × 7B | 33.90 | 24.57 | 24.48 | 24.52 | _17.37_ | 15.68 | 15.68 | 15.68 | _24.16_ | 18.28 | 9.78 | 9.78 | 9.78 |
| *Training-based* | | | | | | | | | | | | | | |
| Self-RAG (LLaMA-2) | 7B | 29.96 | 67.82 | 66.97 | 67.39 | 6.90 | 22.34 | 32.40 | 26.45 | 2.34 | 1.98 | 10.53 | 18.80 | 13.50 |
| | 13B | 31.66 | 71.26 | 70.35 | 70.80 | 6.07 | 30.46 | 40.20 | 34.66 | 1.90 | 1.33 | 12.79 | 20.90 | 15.86 |
| FRONT (LLaMA-2) | 7B | 40.84 | 77.70 | 69.89 | 73.59 | 9.18 | 58.60 | 55.33 | 56.92 | 11.50 | 21.38 | _24.74_ | 24.84 | _24.79_ |
| | 13B | _41.51_ | 78.44 | 73.66 | 75.97 | 9.32 | **60.31** | _59.21_ | **59.75** | 11.94 | _22.61_ | **24.86** | _25.39_ | **25.12** |
| Evidence-R1 (LLaMA-2) | 7B | 37.68 | _79.04_ | _78.61_ | _78.82_ | 10.03 | 53.35 | 58.48 | 55.80 | 18.88 | 20.53 | 6.82 | **25.79** | 10.79 |
| | 13B | 40.48 | **81.98** | **82.33** | **82.15** | 11.33 | _59.33_ | **60.06** | _59.69_ | 20.72 | 21.41 | 6.40 | 23.62 | 10.07 |

tation quality on the ASQA dataset and in citation precision on the ELI5 and QAMPARI datasets, even surpassing ChatGPT. The reason is that conducting interpretable and explicit reasoning at the sentence-level evidence prior to citation generation yields more rational citations. Additionally, subsequent filtering of the cited sentence-level evidence via the internal self-checking mechanisms of LLMs further mitigates citation errors effectively. However, in terms of citation recall, performance exhibits a decreasing trend across the ASQA, ELI5, and QAMPARI datasets. The proposed method prioritizes the retrieval of sentence-level evidence with the highest relevance, aiming to mitigate the risk of hallucination caused by incorrect citations. In contrast, the ELI5 dataset demands the integration of information across multiple passages to generate long-form answers, while QAMPARI, which requires answers in the form of comma-separated entity lists derived from diverse passages, generally contains evidence with lower relevance. These characteristics collectively increase the difficulty of retrieving all related evidence, thereby contributing to the observed performance decline, especially particularly for our method under the 13B setting.

In conclusion, results in Table 1 can be attributed to the fact that demonstrate that Evidence-R1 not only matches the citation precision of existing state-of-the-art models but also achieves significant improvements in interpretability and traceability compared to Self-RAG and FRONT.

## 4.4 ANALYSIS STUDY

We conduct ablation studies to verify the effectiveness of different components proposed in Evidence-R1.

### 4.4.1 ABLATION STUDY ON DIFFERENT TRAINING STAGES AND COMPONENTS.

To demonstrate that the two-stage training, Explicit Reasoning and Implicit Reflection can boost the performance of Evidence-R1, we conduct a series of ablation studies. Table 2 gives the ablation results. Compared with "w/o MRDAA" which means cutting down the Multi-reward Dependence-aware Alignment and fine tune only with Supervised Fine-tuning, Evidence-R1 obtains more citation quality over all the datasets. However, Compared with "w/o MRDAA and ER" and "w/o MRDAA and IR", which only train the Implicit Reflection and Explicit Reasoning process with Supervised Fine-tuning, "w/o MRDAA" is not more effective than any one of them. This is because the percentage asymmetry in the sentence-level evidence they rely on undermines the model's performance, which further demonstrates the importance of MRDAA. Additionally, compared with the setting of "w/o MRDAA, ER, IR" (VANILLA-SFT, coming from FRONT) which is required to

directly generate answers with citations via supervised fine-tuning, the model equipped with each of these components achieves significant improvements. This validates the effectiveness of our implementation of Explicit Reasoning or Implicit Reflection.

Table 2: Ablation study on different training stages and components.

| Models | ASQA | | | | ELI5 | | | | QAMPARI | | | | |
|---|---|---|---|---|---|---|---|---|---|---|---|---|---|
| | Correctness | Citation | | | Correctness | Citation | | | Correctness | | Citation | | |
| | EM Rec. | Rec. | Prec. | F1 | Claim | Rec. | Prec. | F1 | Rec.-5 | Prec. | Rec. | Prec. | F1 |
| Evidence-R1-7B | 37.68 | 79.04 | 78.61 | 78.82 | 10.03 | 53.35 | 58.48 | 55.80 | 18.88 | 20.53 | 6.82 | 25.79 | 10.79 |
| {- MRDAA} | 39.56 | 76.28 | 77.38 | 76.83 | 10.4 | 51.85 | 53.91 | 52.86 | 18.48 | 19.75 | 6.25 | 23.23 | 9.85 |
| {- MRDAA, ER} | 38.78 | 78.21 | 78.05 | 78.13 | 10.13 | 52.87 | 56.73 | 54.73 | 17.22 | 18.98 | 5.71 | 21.64 | 9.04 |
| {- MRDAA, IR} | 39.42 | 77.09 | 78.41 | 77.74 | 10.8 | 54.50 | 55.47 | 54.98 | 18.82 | 20.63 | 18.38 | 18.60 | 18.49 |
| {- MRDAA, ER, IR} | 40.32 | 67.67 | 63.67 | 65.61 | 9.63 | 42.30 | 40.06 | 41.15 | 12.86 | 21.09 | 21.35 | 21.36 | 21.35 |

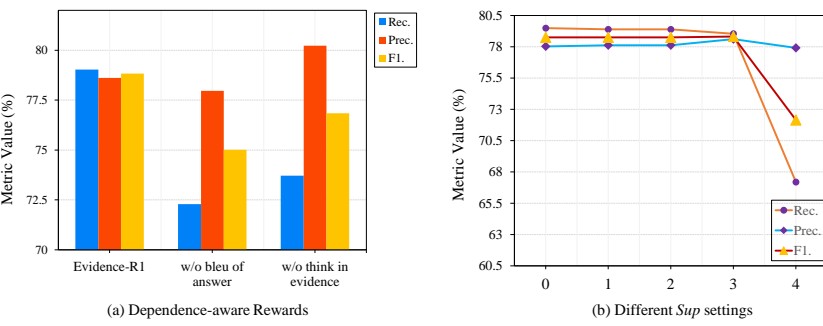

(a) Dependence-aware Rewards      (b) Different *Sup* settings

Figure 4: Ablation results on Dependence-aware Rewards and different $Sup$ settings with ASQA.

### 4.5 EFFECTS OF DEPENDENCE-AWARE REWARDS AND THE REFLECTION TOKEN

Our method leverages MRDAA with Dependence-aware Rewards to enhance the consistency between explicit reasoning and implicit reflection. As depicted in the left section of Figure 4, the overall citation quality of Evidence-R1 degrades when either the alignment between reflection tokens and answers ("w/o bleu of answer") is removed or the alignment between the "think" section and subsequent citation ("w/o think in evidence") is eliminated. This thus validates the effectiveness of our Dependence-aware Rewards setup. A more detailed comparison can be found in Appendix B.3.

Results in the right section of Figure 4 indicate that as the threshold of Citation Filtering increases, the citation recall of Evidence-R1 decreases continuously while the citation precision increases steadily. Except for the threshold value of 5, its excessive filtering has caused all indicators to decrease. This demonstrates that our method can effectively reduce traceability errors and focus on more precise traceability.

## 5 CONCLUSION

This paper proposes Evidence-R1, a novel RAG generator framework that conducts fine-grained verification for generated in-line citations in RAG by driving explicit reasoning and implicit reflection on sentence-level evidence. By introducing MRDAA, a multi-rule tree reward mechanism, we enhance the consistency between the two processes regarding the sentence-level evidence they rely on through reinforcement learning. Finally, we filter the citations using scores in $Sup$ to ensure high-quality sentence-level citations. Results from experiments on the ALCE benchmark demonstrate the effectiveness of Evidence-R1. Conducting interpretable and explicit reasoning on sentence-level evidence prior to citation generation yields more rational citations. Additionally, subsequent filtering of the sentence-level citations via implicit reflection further mitigates citation errors effectively. Specifically, Evidence-R1 not only outperforms state-of-the-art models (including ChatGPT) in citation precision, but also achieves significant improvements in interpretability and traceability, thereby enhancing RAG explainability.

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

## A USE OF LLM

During the writing of this paper, a Large Language Model (LLM) was used as an auxiliary tool. The specific LLM tool employed is Doubao, and its application scope was strictly limited to grammar checking and simple content polishing, with no involvement in core research links. The detailed roles are explained as follows:

### A.1 GRAMMAR CHECKING

After the first draft of the paper was completed, this LLM was used to verify the grammatical accuracy of the entire text. The focus was on identifying and addressing basic grammatical issues in English expressions, including tense errors, subject-verb agreement mistakes, improper use of articles, and incorrect punctuation. For instance, it helped organize the grammatical logic of long and complex sentences in the abstract and research methodology sections, and corrected grammatical deviations caused by differences between Chinese and English expression habits. However, the authors did not directly adopt the LLM's suggestions for grammatical revisions; all grammatical corrections were manually checked and confirmed by the authors to ensure the accuracy and applicability of the revised results.

### A.2 SIMPLE CONTENT POLISHING

On the basis of grammar checking, the LLM was used to slightly optimize the linguistic fluency of certain parts of the paper, focusing on enhancing the conciseness of sentence structures and the clarity of expressions. Examples include streamlining redundant expressions in the research results analysis section, adjusting the word order of some long sentences to improve reading fluency, and fine-tuning the linguistic rhythm of the research background description in the introduction to ensure smoother content transitions. Importantly, such polishing did not alter the core semantics, research viewpoints, or logical framework of the original text; it only involved minor optimizations based on the original content expressions. All polished content was ultimately reviewed by the authors to ensure that the polished expressions were fully consistent with the research intentions and that no new viewpoints or information were introduced.

### A.3 RESPONSIBILITY STATEMENT

The authors of this paper clearly understand and strictly adhere to the relevant regulations of ICIR regarding the use of LLMs, and assume full responsibility for all content of the paper published under the authors' names. The aforementioned LLM was only used as an auxiliary tool for grammar checking and simple content polishing, and did not participate in any core research work such as research conception, data collection and analysis, or derivation of research conclusions. Its role did not meet the criteria of a contributor. All research content, academic viewpoints, and logical deductions in this paper were independently completed by the authors, ensuring the originality, scientificity, and authenticity of the paper. There is no risk of plagiarism or academic misconduct caused by the use of the LLM.

## B MORE DETAILS OF EXPERIMENTAL SETUP

### B.1 IMPLEMENTATION DETAILS

We provided the top 5 retrieved documents to serve as context for each question and employed ChatGPT to automatically synthesize high-quality training data. During the two Stage Training, we trained the LLaMA-2 series (7B and 13B) for 5 epochs with a learning rate of 2e-5 through supervised fine-tuning (SFT). And for Group Relative Policy Optimization (GRPO) of MRDAA, we implemented a sample selection step to prioritize high-quality data that included at least one $Sup$ greater than 2. This filtering ensured that the GRPO update was driven by informative high-signal samples, mitigating the impact of noisy or low-quality data. All experiments were carried out on NVIDIA A100 * 8 80G GPUs.

### B.2 STATISTICS FOR DATA GENERATION

For ease of comparative analysis, we use the same original dataset as FRONT for data synthesis. Table 3 shows the statistics of the dataset for automatically synthesizing training data. There are 5,667 long-form questions with an average of 69.15 words per answer and 6.94 citations per answer. And there are 2,431 short-form questions, with an average of 4.68 words per answer and 3.77 citations per answer.

During the data generation, we employed specific steps as follows:

Table 3: Statistics of the dataset for automatically synthesizing training data

| Questions | Number | Avg. Words | Avg. Citation |
|---|---|---|---|
| Long Answer | 5667 | 69.15 | 6.94 |
| Short Answer | 2431 | 4.68 | 3.77 |

- First, based on the given question and the top 5 retrieved documents, we prompted Chat-GPT to generate answers as Table 6 and Table 7. Each sentence in the answers should cite relevant passages in the format of [1][2].

- Next, we used NLTK to split the cited relevant passages into sentences, and then employed ChatGPT to generate a sentence-level thinking process to infer the answers, using only sentence labels in the format of "sentence 1_1, 1_2", as shown in Table 8.

- Then, in Table 9, we split the answers into individual sentences and extracted sentence-level evidence for each sentence of the answers.

- Finally, for each piece of sentence-level evidence, we generated the reflection token ($Sup$) based on the given question and the current sentence in the answers, following Table 10. Then we combine the given question, sentences of the top 5 retrieved documents, sentence-level thinking process, answers, sentence-level evidence, and reflection token to generate our structured, high-quality training data, such as Table 11.

### B.3 FULL ABLATION RESULTS ON DEPENDENCE-AWARE REWARD AND THE REFLECTION TOKEN

Table 4: Full Ablation results on Dependence-aware Reward.

| Model Type | ASQA | | | ELI5 | | | QAMPARI | | |
|---|---|---|---|---|---|---|---|---|---|
| | Rec. | Prec. | F1. | Rec. | Prec. | F1 | Rec. | Prec. | F1 |
| Evidence-R1-7b | 79.04 | 78.61 | 78.82 | 53.35 | 58.48 | 55.80 | 6.82 | 25.79 | 10.79 |
| -w/o bleu of answer | 72.29 | 77.97 | 75.02 | 46.45 | 58.15 | 51.65 | 6.32 | 22.81 | 9.90 |
| -w/o think in evidence | 73.72 | 80.23 | 76.84 | 48.27 | 60.53 | 53.71 | 6.15 | 22.67 | 9.68 |

Table 5: Full Ablation results on the reflection token.

| Model Type | ASQA | | | ELI5 | | | QAMPARI | | |
|---|---|---|---|---|---|---|---|---|---|
| | Rec. | Prec. | F1. | Rec. | Prec. | F1 | Rec. | Prec. | F1 |
| Evidence-R1-7b | | | | | | | | | |
| $+Sup > 0$ | 79.49 | 78.03 | 78.75 | 55.07 | 55.94 | 55.50 | 6.82 | 25.79 | 10.79 |
| $+Sup > 1$ | 79.39 | 78.12 | 78.75 | 55.07 | 55.99 | 55.53 | 6.74 | 25.98 | 10.70 |
| $+Sup > 2$ | 79.39 | 78.12 | 78.75 | 55.07 | 55.99 | 55.53 | 6.74 | 25.98 | 10.70 |
| $+Sup > 3$ | 79.04 | 78.61 | 78.82 | 53.35 | 58.48 | 55.80 | 6.66 | 27.83 | 10.75 |
| $+Sup > 4$ | 67.19 | 77.91 | 72.15 | 34.40 | 56.56 | 42.78 | 1.08 | 4.10 | 1.71 |
| Evidence-R1-13b | | | | | | | | | |
| $+Sup > 0$ | 81.98 | 82.33 | 82.15 | 59.33 | 60.06 | 59.69 | 6.35 | 22.91 | 9.94 |
| $+Sup > 1$ | 81.36 | 82.77 | 82.06 | 58.90 | 60.47 | 59.67 | 6.40 | 23.62 | 10.07 |
| $+Sup > 2$ | 81.25 | 82.68 | 81.96 | 58.72 | 60.48 | 59.59 | 6.16 | 23.29 | 9.74 |
| $+Sup > 3$ | 78.64 | 83.11 | 80.81 | 53.07 | 64.10 | 58.07 | 5.16 | 21.83 | 8.35 |
| $+Sup > 4$ | 62.02 | 78.87 | 69.44 | 30.87 | 55.49 | 39.67 | 1.29 | 5.63 | 2.10 |

Table 6: Prompt for Long-form Questions with cited answer generation.

Instruction: Given a question, and a paragraph, Your task is to write an accurate, engaging, and concise answer for the given question using only the provided paragraph (some documents of which might be irrelevant) and cite them properly, such as [5]. Use an unbiased and journalistic tone. Always cite for any factual claim. When citing several search documents, use [1][2][3]. Cite at least one document and at most three documents in each sentence. If multiple documents support the sentence, only cite a minimum sufficient subset of the documents.
The detailed criterion is as follows:
The requirement is to return strictly in JSON format and only output JSON. Format: {"Answer":""} , where the "Answer" is the right answer of given question based on the given paragraph. If there is no relevant document to answer the question, you should generate the best answer that you think. For each sentence of the answer, you should cite the relevant documents properly, such as [2][5].
For Example:
## Question: I need to organize a picnic for a large group of people. How can I do that?
## Paragraph:
Document [1]: Picniclarge picnics ...
Document [2]: Some picnics ...
...
Document [5]: ...
## Output:
{"Answer": "To organize a picnic for a large group in established ... is not a cookout[3]."}

## Question: {Question}
## Paragraph: {Documens}
## Output:

Table 7: Prompt for short-form Questions with cited answer generation.

Instruction: Given a question, and a paragraph, Your task is to provide a list of accurate answers for the given question using only the provided paragraph (some documents of which might be irrelevant) and cite them properly, such as [2][5].
Always Cite at least one document for each answer and When citing several documents, use [1][2][3]. Separate answers by commas. For question that have more than 5 answers, write at least 5 answers. The detailed criterion is as follows:
The requirement is to return strictly in JSON format and only output JSON. Format: {"Answer":""}, where the "Answer" is the right answer of given question based on the given paragraph. If thereis no relevant document to answer the question, you should generate the best answer that you think.
For Example:
## Question: who sings knock three times on the ceiling if you want me?
## Paragraph:
Document [1]: Picniclarge picnics ...
Document [2]: Some picnics ...
...
Document [5]: ...
## Output:
{"Answer": "Irwin Levine [1], L. Russell Brown [1][3]."}

## Question: {Question}
## Paragraph: {Documens}
## Output:

Table 8: Prompt for sentence-level thinking process.

Instruction: Given a question, a paragraph and an answer of the question, Your task
is to provide a reasoning process based on sentence-level evidences in the paragraph
in order to infer the answer.The paragraph consists of many sentences,
and each sentence has an id, such as "[2_1]".
The detailed criterion is as follows:
The requirement is to return strictly in JSON format and only output JSON. Format:
{"Reason":""}, where the "Reason" is the reasoning process how we can reason
to the answer based on sentence-level evidences in the paragraph. If there no
relevant information in the paragraph , you should output the best reasoning
process that you think. When referring to the content in relevant sentences, you should
use the sentence ID, such as "[2_1]". When there are several sentence-level
evidences that can be used to infer the answer, all of them should be included
in the reasoning process. If multiple evidences support the answer, give a maximum
sufficient subset of the evidences.
You must output the JSON only like: {"Reason":""}
For Example:
##Question: I need to organize a picnic for a large group of people. How can I do that?
##Paragraph: [1_1]picnics.[1_2]In ... all to share.[3_1]When the picnic is not also
a cookout, the food eaten is rarely ...[3_2]The first usage of the word is traced to
##Answer: To organize a ... soft drinks available.
##Output:
{"Reason": "To organize a picnic ...[3_2]"}

##Question: {Question}
##Paragraph: {Paragraph}
##Answer: {Answer}
##Output:

Table 9: Prompt for extracting sentence-level evidence.

Instruction: Given an question, a paragraph and an answer of the question. Your task is to
extract sentence-level evidences in order to infer the answer based on the information in
the paragraph. The paragraph consists of many sentences, and each sentence has an id,
such as "[2_1]". When there are several sentence-level evidences, use ["2_1","2_16","5_1"]
and you must output the ids only.
The detailed criterion is as follows:
The requirement is to return strictly in JSON format and only output JSON.
Format: {"Evidences":[]}, where the "Evidences" is a list of the sentences ids , you must
output the ids only. If there is no relevant information in the paragraph, then the
"Evidences" should be "[]".
For Example:
##Question: I need to organize a picnic for a large group of people. How can I do that?
##Answer": You can also consider ...'
##Paragraph: [1_1]Picniclarge picnics [2_2]In ... , and restrooms.[3_1]Some ...
tertainment at which each person ... drinks.[5_2]The first usage of the word is traced to
##Output:
{"Evidences":["3_1"]}

##Question: {Question}
##Answer: {Answer}
##Paragraph: {Paragraph}
##Output:

Table 10: Prompt for the reflection token.

Instruction: Given an question, an answer of the question and an evidence of the answer.
Your task is to evaluate if the answer is fully supported by the information provided
in the evidence. The detailed criterion is as follows:
The requirement is to return strictly in JSON format and only output JSON. Do not
include the relevant analysis process.
Format: {"IsSup":4}.
Use the following entailment scale to generate a score for "IsSup":
5: Fully supported - All information in Answer is supported by the evidence, or
extractions from the evidence. This is a somewhat extreme case and is only applicable
when the Answer and part of the evidence are almost identical.
4: Mostly supported - Most of the information in the Answer is supported by the evidence,
but there is some minor information that is not supported. In other words, if an Answer
is a paraphrase of the evidence or a less concrete version of the descriptions of the evidence,
it should be considered a 4.
3: Partially supported - The Answer is supported by the evidence to some extent, but there
is major information in the Answer that is not discussed in the evidence. For example, if
an question asks about two concepts and the evidence only discusses either of them, it
should be considered a 3. If the Answer covers a lot of new information that is not discussed
in the evidence, it should be 3.
2: Little support - The Answer and evidence are only loosely related, andmost of the
information in the Answer isn't supported by the evidence.
1: Ignore / Contradictory - The Answer completely ignores evidence or contradicts the
evidence. This can also happen if the evidence is irrelevant to the question.
Make sure to not use any external information/knowledge to judge whether the Answer is
true or not. Only check whether the Answer is supported by the evidence, and not
whether the Answer follows the question or not.
You must output the JSON only, and can not output the text in $< think >$ and $< /think >$
For Example:
##Question: I need to organize a picnic for a large group of people. How can I do that?
##Answer": You can also consider making it a potluck where each person contributes
some dish to a common table for all to share.
##Evidence: [4_1]When the picnic is not also a cookout, the food eaten is rarely hot,
instead taking the form of deli sandwiches, finger food, fresh fruit, salad, cold meats and
accompanied by chilled wine or champagne or soft drinks
##Evaluate:
{"IsSup":1}
##Question: I need to organize a picnic for a large group of people. How can I do that?
##Answer": To organize a picnic for a large group in established public parks,
you can choose a picnic area that generally includes picnic tables and possibly other items
related to eating outdoors, such as built-in grills, water faucets, garbage containers,
and restrooms. ##Evidence: [2_2]In established public parks, a picnic area generally
includes picnic tables and possibly other items related to eating outdoors, such as
built-in grills, water faucets, garbage containers, and restrooms
##Evaluate:
{"IsSup":5}
##Question: {Question}
##Answer: {Answer}
##Evidence: {Paragraph}
##Evaluate:

Table 11: A case of our structured, high-quality training data.

"Instruction": "Given an instruction and a paragraph, your task is to provide a reasoning process, guide answer generation, cite sentence-level evidence from the paragraph and evaluate it's support for the answer.
Paragraphs:[1_1]People in the state senate serve how many years house and senate? [1_2]The House and Senate both have different requirements for being members. [1_3]The House requires its members to be at least 25 years old and US citizens for at least seven years. [1_4]The Senate requires its members to be at least 30 years old and a US citizen for at least 9 years. [1_5]There are also fewer Senators than there are members of the House (100 members of the Senate vs 435 in the House). [1_6]What is the Senators term length in the Senate? [1_7]Senators serve six years. [1_8]What ways is the house closer to the people [2_1]What is a Senator and representative residency requirement elected as a senator? [2_2]Do presidents have to serve one term as senator? [2_3]How long is the term for a state representative in Iowa? [2_4]It is two years for a representative and four years for a state senator. [3_1]Just another WordPress site states before they can be in the house of rep. what is the title of the leader of the house of rep speaker of house How many people serve in the united states senate 100 [3_2]How long is the term for a member of the united states senate [4_1] What legislative body is composed of 100 members 535 people. [4_2]Who works in the legislative branch? [4_3]The house of representatives (100) and the senators Which part of congress with 100 members? [5_1]What is the Difference Between the House and the Senate? - [5_2]World Atlas.com What is the Difference Between the House and the Senate? [5_3]The House has 435 members while the Senate has 100 members.
Instruction: what are the requirements to be in the state senate the house"

"output": "$<think>$To be in the state Senate, sentence 1_4 specifies that members must be at least 30 years old and a US citizen for at least 9 years. For the House, sentence 1_3 states the requirements as at least 25 years old and US citizenship for at least seven years. Furthermore, sentence 1_7 mentions that Senators serve six years, and sentence 1_5 highlights that there are fewer Senators (100) compared to members of the House (435), indicating a distinction in the number of members between the two chambers. $</think><statement>$To be in the state Senate, one must be at least 30 years old and a US citizen for at least 9 years, whereas for the House, the requirements are at least 25 years old and US citizenship for at least seven years.[1_3][Supported:3][1_4][Supported:4]$</statement><statement>$Senators serve a term of six years.[1_7][Supported:5]$</statement><statement>$Additionally, there are fewer Senators (100) compared to members of the House (435).[1_5][Supported: 5]$</statement>$"

