# OpenReview forum: "Evidence-R1: Fine-Grained Evidence-Driven Explicit Reasoning and Implicit Reflection for Enhancing RAG Explainability via Reinforcement Learning"
_ICLR.cc/2026/Conference — ICLR 2026 Conference Withdrawn Submission_

### Official Review · Reviewer_4d6f · 2025-10-19

**Soundness:** 3
**Presentation:** 3
**Contribution:** 2
**Rating:** 6
**Confidence:** 4

**Summary:**

The paper proposes Evidence-R1, a RAG generation framework that enforces fine-grained, sentence-level attribution through a dual-process design: (i) explicit reasoning (Evidence-of-Thought, EoT) that must cite sentence-level evidence while deriving each step, and (ii) implicit reflection that rates, via a special token Sup on a 1–5 scale, how well each answer statement is supported by the cited sentences. The authors build an automatic data synthesis pipeline (using ChatGPT) to create structured training targets containing <think> reasoning, <statement> answers with sentence-level citations, and Sup scores.

**Strengths:**

1. The method is practical, pairing “show your work” reasoning (EoT) with a per-sentence support score (Sup) so the model both cites and self-checks.
2. The outputs are tightly specified (sentence IDs and Sup per sentence), which makes auditing and automated filtering straightforward.
3. On ALCE, the approach shows consistent gains in citation precision against strong baselines while keeping overall answer quality competitive.

**Weaknesses:**

1. The supervision is predominantly synthetic, so the generated reasoning traces, citations, and Sup scores may be noisy, and the paper provides limited human validation or calibration to quantify this noise.
2. The RL objective downweights answer correctness to emphasize citation quality, which can yield well-cited but factually incorrect statements, and the paper does not thoroughly analyze this trade-off.
3. The evaluation is narrow—focused mainly on ALCE—so it remains unclear how the method generalizes to multi-hop questions, long-context settings, or domains with noisier retrieval.
4. The related-work coverage omits several relevant recent approaches on sentence-level attribution, faithful rationales, and verifiable generation, so the contribution is not fully contextualized by prior art.

**Questions:**

Please see the weakness above

---

### Official Review · Reviewer_HM9z · 2025-10-31

**Soundness:** 2
**Presentation:** 2
**Contribution:** 2
**Rating:** 2
**Confidence:** 4

**Summary:**

The paper proposes Evidence-R1, a RAG generator that first produces sentence-level, evidence-grounded reasoning and then performs an internal reflection step to judge how well each answer sentence is supported, aiming to improve explainability, traceability, and citation precision over existing RAG systems. It further introduces a multi-reward, dependence-aware RL scheme (MRDAA) to align the explicit reasoning and the reflective scoring so that both rely on consistent sentence-level evidence.

**Strengths:**

1. The paper targets a real and current gap in RAG: fine-grained, sentence-level citation verification rather than document-level attribution, and implements it with a clear dual-process design (explicit reasoning + implicit reflection).
2. The reinforcement-learning formulation (MRDAA) tries to couple several interdependent rewards (format, citation correctness, EoT evidence use, reflection quality) and the experiments on ALCE show higher citation precision than strong baselines, even ChatGPT, which is a nontrivial result.

**Weaknesses:**

1. The technical novelty is somewhat incremental with respect to recent fine-grained grounded-citation work (e.g. FRONT, Self-RAG) — the paper largely combines (i) structured, cited reasoning, (ii) reflection tokens, and (iii) RL-based alignment, but the jump over these baselines is more of a careful engineering of signals than a clearly new paradigm.
2. The whole pipeline depends heavily on ChatGPT-style synthetic data (thinking traces, sentence splits, Sup scores); the paper acknowledges that manual annotation is hard but does not really quantify data quality or robustness to noise, so the method may inherit biases from the teacher model.
3. Evaluation is narrow in scope (only ALCE: ASQA, ELI5, QAMPARI), and the paper itself shows that citation recall drops markedly on ELI5/QAMPARI when evidence is more scattered — this makes it unclear how well the approach generalizes to harder, multi-source or tool-augmented RAG settings.
4. The proposed MRDAA tree of rewards looks complex and heuristic, with several weights and conditional rewards (format → citation → EoT → Sup); yet the analysis/ablation is relatively light and does not tell us how sensitive the model is to these design choices or to misalignment between the two processes.

**Questions:**

See weaknesses

---

### Official Review · Reviewer_HvYs · 2025-11-02

**Soundness:** 2
**Presentation:** 3
**Contribution:** 3
**Rating:** 4
**Confidence:** 4

**Summary:**

The authors argue that while current RAG models mitigate hallucinations with in-line citations, these citations are often "coarse-grained" (e.g., document-level) and difficult to verify, leading to a lack of fine-grained traceability and potential citation errors. Hence, the paper proposes Evidence-R1, a novel RAG framework built on a dual-process mechanism:
- Explicit Reasoning: The model first generates an "Evidence-of-Thought" (EoT), a reasoning process enclosed in $\<think\>$ tags that infers the answer from specific sentence-level evidence .

- Implicit Reflection: The model then generates the answer, accompanied by a special "Sup" token, which serves as an internal self-check, evaluating how well the generated statement is supported by the cited evidence on a five-point scale .

To enable model to have this "skills",  the authors use an automatic three-step data generation pipeline using ChatGPT for training data generation. For training, they introduce Multi-reward Dependence-aware Alignment (MRDAA), a multi-rule tree reward mechanism trained with reinforcement learning to enhance the consistency between these two processes .

**Strengths:**

1. Novel design of the framework on the dual-process of "explicit reasoning" (EoT) and "implicit reflection", which is trendy with the recent researcher on exploring LLMs' reasoning ability.

2. The technical contribution on introducing multi-rule tree reward mechanism

**Weaknesses:**

I'm particularly concern about the soundness on the method and experiments conduct:
1. The notable decrease in citation recall especially on the ELI5 and QAMPARI datasets. Although the authors' explanation—that their method prioritizes high-relevance evidence while these datasets require integrating information from many (sometimes lower-relevance) passages —is plausible. However, this is a significant **trade-off**, suggesting the model may excel at precision at the cost of completeness for complex, multi-source queries.

2. The experiments are conducted on LLaMA-2 7B and 13B models. While perfectly valid, it would be interesting to see if these significant gains in fine-grained reasoning and alignment generalize to more recent and capable base models.

3. The data synthesis process heavily relies on ChatGPT, while which version of ChatGPT model is used ia not mentioned in the paper. Are they GPT-3.5-Turbo?
As the data synthesis process can be viewed as an distillation process on the teacher model (ChatGPT in the paper's setting) to the student model on a specific form. It'll be necessary to see the result if the teacher model is directly prompted to execute the design of using  Explicit Reasoning + Implicit Reflection framework to solve the task.

**Questions:**

Missing training-based baseline: Effective Large Language Model Adaptation for Improved Grounding and Citation Generation (Ye at al.)

---

### Official Review · Reviewer_TkPg · 2025-11-03

**Soundness:** 2
**Presentation:** 2
**Contribution:** 1
**Rating:** 2
**Confidence:** 3

**Summary:**

The paper proposes Evidence-R1, a novel Retrieval-Augmented Generation (RAG) framework that aims to improve citation explainability and traceability in large language models (LLMs). It introduces two interacting reasoning processes: Explicit reasoning — the model must reason based solely on sentence-level cited evidence (<think> block); Implicit reflection — an internal self-checking mechanism evaluating how well each answer sentence is supported by evidence via a “Sup” token (scores 1–5). A new Multi-Reward Dependence-Aware Alignment (MRDAA) algorithm, built on Group Relative Policy Optimization (GRPO), enforces consistency between explicit and implicit reasoning. Experiments on the ALCE benchmark (ASQA, ELI5, QAMPARI) show improved citation precision and interpretability compared to FRONT (Huang et al., 2024) and Self-RAG (Asai et al., 2024).

**Strengths:**

1. The authors correctly identify that current RAG models (Self-RAG, FRONT) lack fine-grained citation traceability. Their framing of explicit reasoning + implicit reflection offers a conceptually elegant way to enforce factual grounding and interpretability.

2. Introducing multi-reward tree alignment to manage interdependent supervision signals (format, citation, reflection) is a meaningful contribution beyond standard RLHF or GRPO setups.

**Weaknesses:**

1. The paper builds heavily on FRONT (Huang et al., 2024) and Self-RAG (Asai et al., 2024).
The “explicit + implicit” dual-process idea is compelling, but functionally similar to combining chain-of-thought reasoning and self-reflection already seen in DeepSeek-R1 or Self-RAG. The innovation primarily lies in granularity (sentence-level) rather than a fundamentally new paradigm.
2. As for evaluation, statistical significance tests are missing despite small numerical gaps on some datasets.
3. The MRDAA multi-rule tree reward seems theoretically ad-hoc. The cumulative product of ancestor probabilities (Eq. 7) is neither theoretically justified nor empirically ablated in isolation. A simpler alignment loss (e.g., mutual information or KL regularization) might achieve similar gains.

**Questions:**

Please refer to the weaknesses above.

---

### Note · Authors · 2026-01-06

I have read and agree with the venue's withdrawal policy on behalf of myself and my co-authors.